Changes in precipitation may alter food preference in an ecosystem engineer, the black land crab, Gecarcinus ruricola

McGaw Iain J. 1 ijmcgaw@mun.ca
Van Leeuwen Travis E. 2
Trehern Rebekah H. 3
http://orcid.org/0000-0002-0198-4537 Bates Amanda E. 1
1 Department of Ocean Sciences, Memorial University , St Johns, NL , Canada
2 Cape Eleuthera Institute , Rock Sound, Eleuthera , The Bahamas
3 Department of Biosciences, University of Exeter , Exeter, Devon , UK
Toonen Robert
Electronic publication date: 2019 May 8
Publication date: 2019
Volume: 7
Electronic Location ID: e6818
Received 2018 Sep 18; Accepted 2019 Mar 14
Copyright: © 2019 McGaw et al.
Copyright year: 2019
Copyright holder: McGaw et al.
License: This is an open access article distributed under the terms of the Creative Commons Attribution License, which permits unrestricted use, distribution, reproduction and adaptation in any medium and for any purpose provided that it is properly attributed. For attribution, the original author(s), title, publication source (PeerJ) and either DOI or URL of the article must be cited.
License URL: https://creativecommons.org/licenses/by/4.0/

Keywords: Climate change, Dehydration, Gecarcinus ruricola, Water budget, Feeding, Caribbean

Funding: Natural Sciences and Engineering Research Council Discovery Grant 207112 This work was supported by a Natural Sciences and Engineering Research Council Discovery Grant (207112) to Iain J. McGaw. The funders had no role in study design, data collection and analysis, decision to publish, or preparation of the manuscript.

==============================
Gecarcinid land crabs are ecosystem engineers playing an important role in nutrient recycling and seedling propagation in coastal forests. Given a predicted future decline in precipitation for the Caribbean, the effects of dehydration on feeding preferences of the black land crab Gecarcinus ruricola were investigated. G. ruricola were offered novel food items of lettuce, apple, or herring to test for food choice based on water and nutritional (energetic) content in single and multiple choice experimental designs. The effect of dehydration was incorporated by depriving crabs of water for 0, 4, or 8 days, leading to an average body water loss of 0%, 9%, and 17%, respectively, (crabs survived a body water loss of 23% + 2% and 14–16 days without access to water). The results were consistent between the single and multiple choice experiments: crabs consumed relatively more apple and fish and only small amounts of lettuce. Overall, no selective preferences were observed as a function of dehydration, but crabs did consume less dry food when deprived of water and an overall lower food intake with increasing dehydration levels occurred. The decrease in feeding was likely due to loss of water from the gut resulting in the inability to produce ample digestive juices. Future climatic predictions suggest a 25–50% decline in rainfall in the Caribbean, which may lead to a lower food intake by the crabs, resulting in compromised growth. The subsequent reduction in nutrient recycling highlights possible long-term effects on coastal ecosystems and highlights the importance of future work on climate relative behavioral interactions that influence ecosystem function.

Introduction

Ecological research on climate change has largely focused on the influence of environmental temperature as a driver for changes in biodiversity, nevertheless, global precipitation regimes are also shifting with wet regions receiving increasingly more rainfall and drier regions becoming drier (Donat et al., 2016). Strong evidence suggests that desiccation can challenge water balance in terrestrial organisms, and thus set physiological constraints which in turn limit a species distribution (Terblanche & Overgaard, 2015). By comparison, behavioral changes that allow species to adapt to the new climatic conditions have received less research effort than physiological mechanisms (Bellard et al., 2012). Thus, behavioral flexibility is an additional mechanism that will not only influence species vulnerability to changing climate conditions, but also impact species that play key functional roles within ecosystems (Wong & Candolin, 2015).

The Caribbean region is one of the most vulnerable areas with respect to climate change (Taylor et al., 2018). Not only is this region likely to experience gradual warming with average annual temperatures increasing by 0.6–4 °C by the end of the century (Campbell et al., 2011; Taylor et al., 2018), more importantly this temperature change will be accompanied by a significant change in precipitation levels. At present the majority of rain in the Caribbean falls between May and October, with the dry season starting in November and peaking in February and March (Chen et al., 1997; Campbell et al., 2011). Although specific models vary between the northern and southern Caribbean regions, most predict a drying scenario. Overall rainfall in the Caribbean will decrease by approximately 25%, but this could reach as high as 50% in some regions (Nurse & Sem, 2001; Christensen et al., 2007; Campbell et al., 2011). Although precipitation levels are predicted to decrease, this trend will not be consistent throughout the entire year. The dry season is predicted to become somewhat wetter with an increase in major rainfall days, whereas, the number of dry days in the wet season will increase, especially during the early part (May–July) of the season (Christensen et al., 2007; Campbell et al., 2011; Hall et al., 2013; Taylor et al., 2011, 2013).

Brachyuran crabs of the family Gecarcinidae are large tropical and sub-tropical land crabs and offer a compelling model taxon to investigate the impacts of changing precipitation regimes because they are dependent on access to moisture. The crabs inhabit shaded forests and scrub land where they construct burrows in soft earth or shelter among tree roots (Hartnoll et al., 2006). Land crabs can be found many kilometers from the sea and at altitudes of up to 1,000 m above sea level (Chace & Hobbs, 1969; Britton, Kroh & Golightly, 1982; Jiménez et al., 1994). The family Gecarcinidae contains six genera including crabs within the genus Gecarcinus which range in distribution from subtropical areas of North and South America (Florida to Venezuela) and throughout the Caribbean Islands. The genus Gecarcinus currently includes four species of which the black land crab, Gecarcinus ruricola, is the most terrestrial of the Caribbean land crabs (Taylor & Davies, 1982). Although these crabs are classified as terrestrial they still have to return to the sea to deposit their eggs. The larval stages develop at sea but return to land en masse as megalopae after approximately 1 month (Hartnoll & Clark, 2006).

A major obstacle associated with the movement onto land is water loss; while land crabs are substantially less permeable than their aquatic counterparts, they do not approach the levels of impermeability seen in true terrestrial arthropods. Therefore, water loss by evaporation, primarily across the body surface and in the urine and feces, remains an important stressor (Herreid, 1969; Wolcott, 1992). The ability to tolerate desiccation varies within the family Gecarcinidae as a function of terrestriality. For example, Cardisoma species can tolerate between 15–20% loss of body water (Gifford, 1962; Wood, Boutilier & Randall, 1986; Burggren & McMahon, 1981; Harris & Kormanik, 1981), whereas, G. lateralis tolerates, on average, 21% body water loss, with some individuals losing over 30% of their body water before they succumb (Flemister, 1958; Bliss, 1968). Because of this high potential for water loss, land crabs must have mechanisms to avoid desiccation; they can do this by constructing burrows, hiding in crevices, or becoming semi-dormant and reducing metabolism during periods of drying (Wood, Boutilier & Randall, 1986; Bliss et al., 1978; Wolcott, 1992). The crabs usually retreat to burrows in the winter when the temperature drops below 15–18 °C, plugging the burrow and storing leaves as a food source. Not only does temperature play a part in initiating this behavior, it also helps them avoid water loss during the dry winter period (Bliss et al., 1978). Unlike some of the less terrestrial crab species (e.g., Cardisoma, Ocypode), crabs within the genus Gecarcinus usually do not have access to moisture in the burrow, so they have to reduce their activity to conserve water. They usually only emerge from their burrows after rains or when the humidity is high; this behavior itself may limit growth rates (Bliss et al., 1978).

Like most aquatic crabs, land crabs are classified as opportunistic omnivores because their diet can include carrion, insects, animal feces, and plant material (Fimpel, 1975; Bliss et al., 1978; Wolcott & Wolcott, 1984; Ortega-Rubio et al., 1997). However, the nature of their habitat is such that they are primarily herbivorous, foraging on green leaves, herbaceous plants, flowers, and fleshy fruits, favoring these over dry leaf litter (Herreid, 1963; Wolcott & Wolcott, 1984; Kellman & Delfosse, 1993; Greenaway & Raghaven, 1998; Capistrán-Barradas, Moreno-Casasola & Defeo, 2006). This selective nature may be based on nutritional value, size and/or the chemical composition, for instance, G. lateralis may avoid leaves with a high alkaloid content (Capistrán-Barradas, Moreno-Casasola & Defeo, 2006). Although land crabs can be selective, access to high quality food is limited in many environments and subsequently they are often forced to feed on a poor quality diet that is low in nitrogen and water content (Bliss et al., 1978; Wolcott & Wolcott, 1987; Linton & Greenaway, 2007).

Gecarcinid crabs can reach remarkable densities in some areas and have been described as ecosystem engineers because they are important in nutrient recycling, taking over the role of earthworms (Sherman, 2002; Griffiths, Basma & Vega, 2007; Lindquist et al., 2009). They reduce the amount of surface detritus and their burrowing activity aerates and turns-over the soil. The crabs introduce nutrients deep into soil when they bring food down into the burrows and via the subsequent production of faeces (Kellman & Delfosse, 1993; Sherman, 2003, 2006). Land crabs have also been found to feed selectively on seeds and seedlings which makes them key drivers of tropical forest recruitment (Sherman, 2002; Lindquist et al., 2009). In addition the land crab fishery is important throughout sub-tropical and tropical regions. Land crabs are a major source of protein, economics and subsistence for many Caribbean Islanders (Baine et al., 2007); however, they are susceptible to over harvest (Alayon, 2005; Baine et al., 2007). Given the ecological and socio-economic importance and a future scenario of increased drying of the habitat of Gecarcinus crabs the first aim of the study was determine the levels of water loss that the black land crab, G. ruricola, could tolerate as well as the basic metabolic changes accompanying dehydration. Secondly, we hypothesized that crabs of differing dehydration status would exhibit selective feeding and choose different food items dependent on the water or nutrient (energetic) content of the item being offered (Erickson et al., 2008; Nordhaus, Salewski & Jennerjahn, 2011). Finally, because these crabs play an essential role in nutrient recycling in coastal forests, we discuss how potential changes in feeding patterns could be important when predicting responses to global environmental change for species which are strong community players and influence ecosystem function.

Methods and Materials

Crab collection and housing

Eleuthera Island, The Bahamas is a largely undeveloped island. The limestone base is covered in a thin layer of sand/limestone particle soil which does not retain much water. The coastal forests consist largely of pine (Casuarina equisetifolia) close to the shoreline, which give way to scrub and mixed deciduous forest (Bahamas, National Trust). The majority of the rainfall on south Eleuthera falls between March and August, with a noticeable increase in the number of days without rain occurring between September and December (Ciabatta et al., 2018).

Intermoult adult male and female black land crabs, G. ruricola, of 110–460 g were collected by hand at night, primarily from the mixed deciduous forest (February–May 2017). Crabs were transferred to the Cape Eleuthera Institute where they were housed in a slatted wooden hutch 170 × 170 × 170 cm with cardboard tubes providing a shelter for the crabs. The hutch was located under a shaded awning which maintained temperatures between 20–28 °C and the animals were subjected to a natural day-night cycle. The crabs had free access to shallow plastic trays of fresh and salt water and were fed green leaves (mangrove species and sapodilla) ad lib. Animals were acclimated to these conditions for at least 7 days prior to being used in experiments. The animals were sexed and males and females randomly assigned to treatments. The treatment and care of the G. ruricola complied with both Canadian and Bahamian care protocols for crustaceans. All crabs used in the feeding preference experiments were returned to the site of capture after use.

Responses to dehydration

In an initial series of experiments the crabs (n = 8) were deprived of water to determine the maximal survivable water loss. They were not fed for the duration of the experiment to avoid changes in mass associated with food consumption or production of metabolic water. The crabs were held individually in covered perforated plastic boxes of 18 × 12 × 8 cm depth inside the hutch with a diurnal temperature range of 20–28 °C and a relative humidity >80%, these conditions mimicked the burrow environment (Bliss, 1968). Crabs were weighed daily and water loss was expressed as percentage loss of their initial body mass. The experiment was carried out until each animal had become moribund and unresponsive to touch (these animals could be revived by immersion in a tray of freshwater (one to two cm depth) for 24 h). The experiment was then repeated in the experimental dehydration cages (60 × 60 × 60 cm) in the laboratory at a temperature of 25 ± 2 °C. The crabs (n = 10), were weighed daily and the experiment was terminated before they reached their lethal water loss level or noticeable changes in their responsiveness to handling occurred. This approach allowed accurate determination of experimental dehydration treatment periods that would physiologically stress, but not severely incapacitate the crabs.

Oxygen consumption rates (mg O2 kg h−1) were measured to determine if dehydration had any effect on the metabolic rate of the crabs. To measure oxygen consumption the crabs were introduced into Lock and Lock® airtight plastic boxes (Anaheim, CA, USA) 24 × 17 × 9 cm depth of 2.6 L volume and allowed to settle for 3 h after handling. All experiments were performed during the daylight hours since land crabs become very active during the night exhibiting a substantial increase in nocturnal heart rate (McGaw et al., 2018). Air temperature within each plastic box was maintained at 27 ± 1 °C. For readings the lids were sealed and the boxes were covered in black plastic sheeting to avoid visual disturbance to the animals. The boxes remained sealed for 45–70 min which allowed a measurable drop in oxygen without exposing individuals to a hypoxic regime. A 60 ml syringe with a 16 gauge needle was used to collect an air sample. The needle was inserted through a small hole in the lid that was sealed with dental wax. The syringe was pumped in and out three times to circulate the air in the chamber before withdrawing an air sample. The sample was injected through a drierite® column (to remove any moisture) into a Q-S102 O2 analyzer (Qubit Systems, Ontario, Canada). The oxygen analyzer was pre-calibrated with room air as 100% oxygen saturation (20.95% oxygen), and nitrogen gas was used for 0% saturation. The chamber was opened between readings to allow fresh air to circulate. Aerial oxygen consumption (ml kg h−1) was calculated taking into account the volume of the chamber minus the volume of air displaced by the crab in the chamber, the mass of the crab and the length of time the chamber remained closed. This value was converted from milliliters h−1 to milligrams h−1 by multiplying by 1.43 (32 g mol−1 divided by 22.4 l mol−1).

The oxygen consumption of crabs was monitored during an 8 days dehydration period, this was based upon the water loss and survival experiments (described above). The crabs (n = 8) had been starved for 2 days prior to the initial reading (0 day dehydration) because feeding and digestion is associated with an increased metabolic rate termed the specific dynamic action (McGaw, 2005; Secor, 2009). Readings were taken at 0, 2, 4, 6, and 8 days of dehydration during which time the crabs were not fed. Following this 8 days dehydration period the crabs were allowed to rehydrate and oxygen consumption was measured after a 24 h recovery period. A second group of crabs (n = 8) was also monitored under the same time regime, however, this group was given free access to water. This enabled us to determine if changes in oxygen consumption were associated with dehydration as opposed to food deprivation (Ansell, 1973).

The metabolic scope of dehydrated crabs (n = 8) was also calculated using separate animals (from above) that were deprived of water for 0, 4, or 8 days. The crabs were placed in the chambers and allowed to settle for 3 h before a reading was taken; this was the resting metabolic rate (RMR). The crabs were then removed from the chamber and forced to walk for approximately 5 min by constantly agitating them with a stick. A thick elastic band was then wrapped around each side of the carapace and a metal weight was inserted into the bands on the upper surface of the carapace after which the crabs were placed back into the chamber in an inverted position which caused them to struggle vigorously trying to right themselves. This forced activity and subsequent struggling behavior produced the maximal metabolic rate (MMR) (McGaw, 2007). The difference between the RMR and MMR was calculated as the metabolic scope.

Food preferences

Prior to experimentation the crabs were transferred to wire mesh cages (60 × 60 × 60 and 2.5 cm mesh) in the laboratory and deprived of water for 0, 4, or 8 days. This represented a water loss of approximately 0%, 9%, and 17% of the body mass, respectively. The feeding regime was also controlled during this time so at the time of experimentation the crabs had been fasted for 8 days for each dehydration level. A fasting period of 8 days was selected because crabs produced faeces for up to 6 days after consuming large meals (I. McGaw, 2017, personal observation); this period also ensured the stomach was empty and they would feed when offered food (Mchenga & Tsuchiya, 2010).

To determine food preferences individual crabs were held in covered opaque plastic containers (30 × 30 × 60 cm depth) in the laboratory at a temperature of 25 ± 2 °C. The crabs were allowed to settle in the containers for 1 h after handling before weighed portions of the food were introduced. As land crabs exhibit nocturnal foraging behavior (Palmer, 1971) the food was placed in the containers in the evening (approximately 8 pm) and they were left to feed for 12 h; all experiments were carried out in constant darkness. In the morning food was weighed for post-consumption mass. Three different types of food were offered—lettuce leaves (water content = 93.83% ± 0.27%, energetic content = 59 kJ/100 g), apple slices (water content = 85.57% ± 0.56%, energetic content = 218 kJ/100 g), and herring (fish) fillets (water content = 64.88% ± 0.62%, energetic content = 661 kJ/100 g). These items were chosen as novel items that the crabs would not normally encounter to try and ensure the crabs would make a choice based upon water or nutrient (energetic) content of the food. While naturally occurring plants could have been used they did not exhibit pronounced differences in nutrient and water content—more importantly if preference did occur we would be unable to determine if this was affected by familiarity with, or preference for, that naturally occurring item (Thacker, 1996, 1998).

In the first series of experiments, the crabs (n = 14 per food type, and 120–405 g range) were offered just one food item—fresh (unaltered, raw) lettuce, fresh apple, fresh fish, or dry lettuce, dry apple, or dry fish (that had been dried to constant weight in a drying oven at 60 °C). A total of 14 animals (one crab per container) were run at any one time with food types and dehydration levels randomly assigned. This enabled us to determine differences in palatability and feeding rates on each of the foods (Peterson & Renaud, 1989). Because offering single items are not true preference experiments a second experiment was carried out and the crabs were offered a multiple choice of the food items (Peterson & Renaud, 1989; Bergamino & Richoux, 2015). The results of the first series of experiments suggested that the crabs did not eat the dried items as readily as the fresh items. Therefore only the three fresh foods were given to the crabs and they were allowed to feed for 12 h in constant darkness. The three food types were introduced at the same time and an excess of each type was added to ensure the crabs did not consume all of one type and then start feeding on the next type simply because they had consumed all the preferred food items.

In a final series of experiments, a wider size range of crabs (26–475 g) was used to determine if there was any food preference based upon the size of and/or sex animal. For this experiment, only fully hydrated crabs were used and they were only offered the multiple choice of three fresh foods.

Calibration of amount eaten

To control for weight changes of both the fresh and dry food, samples of different shape and mass (n = 22–38) were placed in containers without crabs and weighed again after 12 h. Regression lines were produced for each food type (Table 1) and correction factors were applied to calculate the final mass eaten. Because of the different water content of the three food types and differences in water content between the fresh and dry foods (Table 1), the mass eaten was converted to a dry mass for all food types. Samples of fresh food (n = 18–24) were weighed and dried to constant mass in a drying oven at 60 °C, regression equations used to convert the fresh mass eaten into dry mass eaten (Table 1). The crabs varied in size (carapace width) and even crabs of a similar size varied in mass because of their dehydration status. Therefore, in order to standardize for crab size and wet body mass, the dry body mass of the crab was used. Hydrated crabs (n = 18) varying between 90 and 450 g were weighed and then euthanized by being placed in iced water for 1 h. The crabs were then dried to constant mass in an oven (Table 1). The amount of food eaten was expressed as a dry mass as a percentage of the dry body mass of an individual crab (Steinke, Rajh & Holland, 1993).

Table 1 Changes in food mass.

Item	# Samples	Regression statistics	Equation (start = grams)	R2	
Fresh lettuce—weight change	35	(F = 8,994, p < 0.001)	Final = −0.499 + (0.891 * start)	0.996	
Dry lettuce—weight change	28	(F = 4,121, p < 0.001)	Final = −0.0127 + (1.155 * start)	0.993	
Fresh apple—weight change	38	(F = 12,096, p < 0.001)	Final = −0.110 + (0.927 * start)	0.997	
Dry apple—weight change	31	(F = 12,206, p < 0.001)	Final = 0.144 + (1.038 * start)	0.998	
Fresh fish—weight change	25	(F = 45,089, p < 0.001)	Final = 0.0486 + (1.010 * start)	0.999	
Dry fish—weight change	22	(F = 185,751, p < 0.001)	Final = −0.374 + (0.957 * start)	1.000	
Lettuce—water content	24	(F = 1,410, p < 0.001)	Dry = 0.0508 + (0.0530 * wet)	0.986	
Apple—water content	23	(F = 260, p < 0.001)	Dry = 0.0164 + (0.144 * wet)	0.927	
Fish—water content	22	(F = 1,023, p < 0.001)	Dry = −0.0973 + (0.360 * wet)	0.982	
Crab—water content	18	(F = 744, p < 0.001)	Dry = 3.622 + (0.319 * wet)	0.976	
Note:

Regression statistics and equations for changes in mass of the three fresh and dry food types after 12 h in air at 25 + 2 °C. These were used to calculate the mass eaten and for converting all masses eaten to a dry mass.

Statistical analysis

Cumulative days without rainfall were calculated using the global scale rainfall product, SM2RAIN-CCI (Ciabatta et al., 2018). Rainfall data for the 0.25° grid cell encompassing the Cape Eleuthera Institute (24°49′45″N, 76°19′46″W) was extracted for the time span from January 1, 1998 to December 31, 2015 and quantified cumulative daily rainfall. For each month of each year the total number of consecutive days without rainfall were calculated and the maximum span of days without rainfall for each month was used as the response of interest. This allowed us to calculate historical mean number of days without rainfall, and compare this to chosen times for dehydration (0, 4, 8 days) used in experiments.

Differences between oxygen consumption rates of hydrated and dehydrated crabs as a function of time were tested for using two-way repeated measures ANOVA. Data showing significant effects were further analyzed using Tukey post hoc tests. Differences in MMR and scope were tested for with a one-way repeated measures ANOVA, followed by Tukey post hoc tests to determine where significant differences occurred.

We tested for a preference for certain types of food when the crabs were offered (a) single food items of fresh or dry food and (b) a multiple choice of three types of fresh food, and if (c) crab size (juvenile to adult) had an effect on food selection. For the single food offerings and multiple-choice experiments, the percent body mass food consumed fit the assumptions of a Poisson distribution, and the data was multiplied by 100 and rounded for input into general linear regression models. Body size and sex were included as covariates, but these variables were excluded in model reduction because their inclusion did not reduce the model AIC score (Akaike Information Criterion).

We used a (glm, function glm in the base stats package in R; R Development Core Team, 2017) to test for difference in the quantities of food for crabs offered a single choice. In the multiple-choice experiment crab identification number was initially included as a random effect to account for repeated measures on the same individuals (each crab potentially feeding on the three different food items), using the function glmmPQL in the package MASS (Venables & Ripley, 2002). The random effect of crab identity was not retained because it explained <0.001% of the model variance and inclusion of this parameter did not improve the model fit based on AIC. Thus, we used a glm. To test for food consumption in relation to body mass a generalized least squares regression model using the function gls in the package nlme was performed (Pinheiro et al., 2017). This allowed us to model the unequal variance structure in the different food treatments (lettuce, apple, and fish) using the weights parameter and varIdent.

Model results summary tables report the coefficients for each factor, based on p-values and whether the 95% confidence intervals cross zero. Coefficients represent treatment contrasts of food types apple and fish vs lettuce, moisture level (dry vs fresh food), and days of dehydration exposure 4 and 8, vs 0. The coefficients were used to calculate the % difference in food consumed using the function predict. In each model, fresh lettuce at 0 day dehydration was used as the reference with which to compare the other food and dehydration treatments, because fresh lettuce is similar to naturally occurring food items (leaves) of Gecarcinus species (Bliss, 1968; Wolcott & Wolcott, 1984). Moreover, preliminary experiments using 0 day dehydrated crabs (n = 10) fed green sapodilla leaves indicated no significant difference in the amount of fresh lettuce and green sapodilla leaves consumed by the crabs (Wilcoxon rank sum test with continuity correction; W = 28, p-value = 0.306). This result was supported by a generalized linear model (glm) with crab size as a covariate (results not presented).

Results

Field observations

On Eleuthera Island, G. ruricola, was primarily found in the deciduous forest and scrubland, and was less common in the pine forest closer to the shore. The available food items in the deciduous forest and scrubland were primarily fallen dry leaves, herbaceous plants, and grasses. The crabs were nocturnal, starting to emerge at dusk, retreating to shelter before sunrise. Crabs were only occasionally seen on the surface during the months of December through February but could be collected by excavating burrows or lifting rocks and logs. The animals started to appear on the surface during March and April and were found in large numbers, especially after rains, from mid-April onward. Numerous small burrows were found in the scrubland and under the forest canopy. When we excavated the burrows most were between 30 and 45 cm in length and housed a single small crab (<80 g). The surface soil as well as that at the base of the burrow did not retain any moisture and the dry soil could be easily crumbled between the fingers. Larger burrow entrances were less common and we tended to find larger crabs (>200 g) under rocks and logs, in limestone crevices, or in depressions covered by leaf litter. Due to the porous nature of the soil and bedrock, permanent bodies of standing freshwater were rare. Dew did form overnight during the cooler months (November–April), however, this was less consistent during the remainder of the year, and dews was only evident in open areas on grasses and low lying shrubs. Small pools of standing water persisted for 1–2 days following heavy rainfall. After such events the crabs emerged from the forest en masse (approximately two to seven crabs per m2) and were out in the open during the daylight hours. The crabs gathered in large numbers around these pools to drink water (Fig. 1).

Figure 1 Photo of crabs drinking at a pool.

Black land crabs, Gecarcinus ruricola, emerged after rains in large numbers. This was the only time they were observed in the open during daylight hours. The crabs congregated around standing pools of freshwater and were observed drinking by scooping water with the chelae (photograph—Iain McGaw).

Precipitation levels and responses of crabs to dehydration

The consecutive number of days without rain for each month was plotted for the period 1998–2015 (Fig. 2). There was considerable variation from year to year, however during the month of June, in 4 out of the 18 years, rain fell every day. In contrast, during the months of September through December there were times (between 1 and 4 years) when rain did not fall during the entire month. In general the number of days without rainfall (median levels of 9–12 days) in the months of September–January were similar to one another, but higher than the number of days without rainfall between February and August (median levels of 3 to 7 days), which were similar to one another (one-way ANOVA, df = 11, F = 7.02, p < 0.001).

Figure 2 Rainfall data.

Boxplot of the number of consecutive days per month without rain in a 0.25° grid surrounding the Cape Eleuthera Institute for the years 1998–2015 inclusive. Data was gathered from the new global scale rainfall product, SM2RAIN-CCI. Mean levels for each month are shown as a solid square and the open circles are statistical outliers (values either greater than upper or lower quartile + 1.5 * interquartile difference).

In the water loss experiments, the crabs exhibited a relatively constant daily water loss of between 1.4% and 2% of their body weight (BW) (Fig. 3). The animals became moribund and unresponsive to touch between 14 and 16 days; the mean estimated “lethal” level was 23.7% ± 2.9% body water loss. In the open cages in the lab the rate of water loss was slightly faster (Fig. 3). The animals were maintained for 9 days at which time mean water loss was 19.2%. We thus selected dehydration periods of 4 days and 9.2% ± 0.4% and 8 days and 17.3% ± 1.1% water loss, a regime which ensured that the crabs were not so severely incapacitated that they could not feed or function properly.

Figure 3 Water loss in land crabs.

Water loss (expressed as percent body mass loss) of black land crabs G. ruricola held in perforated plastic containers inside the crab hutch (solid lines, n = 8) and in wire mesh containers in the laboratory (dashed line, n = 10). The former treatment was designed to mimic the burrow environment of the crabs, and animals were maintained in these conditions until all had succumb from water loss. The data represent the mean + SEM.

The oxygen consumption rates were somewhat variable for both dehydrated and hydrated crabs (Fig. 4). There was a significant decline in oxygen consumption of the dehydrated crabs at 8 days (two-way RM ANOVA, df = 15, Interaction, F = 2.96, p = 0.018), whereas, oxygen consumption rates for hydrated crabs remained unchanged during the 8 days treatment and the recovery period. Pre-treatment oxygen consumption rates were regained in the dehydrated crabs within 24 h of rehydration. The MMR of dehydrated crabs (MMR) ranged between mean values of 139 ± 13 and 199 ± 18 mg O2 kg h−1 (Table 2). There was a slight, but significant difference among these values (one-way RM ANOVA, df = 3, F = 2.9, p = 0.048). This occurred because oxygen consumption rates at 4 days were higher than those measured at 8 days and during the recovery period. The metabolic scope varied between 2.4 and 4.6 (Table 2). The metabolic scope of 4.6 measured after 8 days dehydration was significantly higher than that measured at 0 day and after the 24 h recovery period (one-way RM ANOVA on ranks, df = 3, H = 18.22, p < 0.001).

Figure 4 Oxygen consumption rates with dehydration.

Resting oxygen consumption rates (mg O2 kg h−1) of eight hydrated and eight dehydrated black land crabs G. ruricola over a period of 8 days, followed by 1 day of recovery with free access to water. The data represent the mean + SEM, asterisks denote significant differences between the hydrated and dehydrated crabs (p < 0.05).

Table 2 Maximal metabolic rate of crabs.

	0 day	4 days	8 days	R	
Maximal metabolic rate (MMR) (mg O2 kg h−1)	160.2 ± 13.5ab	198.7 ± 17.5b	155.5 ± 13.8a	139.5 ± 12.7a	
Scope	2.5 ± 0.2a	3.2 ± 0.2ab	4.6 ± 0.6b	2.4 ± 0.2a	
Notes:

Maximal metabolic rate (mg O2 kg h−1) and the scope of the response (maximal metabolic rate/resting metabolic rate) of land crabs following 0, 4, and 8 days of dehydration followed by recovery, R after 1 day access to water. The values represent the mean + SEM of eight animals.

Different letters denote significant differences at p < 0.05.

Feeding preferences

When offered single items of fresh or dry lettuce, apple, or fish there was a strong effect of moisture content of the food with animals eating anywhere from three to six times more fresh food than dry food (GLM, df = 238, t = −33.34, p < 0.001; Fig. 5; Table 3). This is because nearly all the animals fed on the fresh food, but less crabs overall fed on the dry food, and the proportion of crabs feeding on the dry food declined with increasing dehydration levels (Table 4). In particular for the dry treatment, there was a significant overall effect of dehydration on feeding; crabs ate less lettuce at both 4 and 8 days, compared to 0 day (GLM, df = 238, t = −16.66 and −20.99, p < 0.001; Fig 5; Table 3). The crabs ate most apple, followed by fish, and consumed significantly less lettuce. In the dry treatment the amount of each food type consumed was not affected by the moisture content of the food, or the number of days the animals had been dehydrated (Fig. 5). However, crabs did consume more fish at 4 days dehydration in the fresh treatment (GLM, df = 238, t = 26.96, p < 0.001).

Figure 5 Feeding on wet and dry foods.

Boxplots showing amount of (A) fresh lettuce, apple, or fish and (B) dry lettuce, apple, and fish (% dry mass as a function of animal dry mass) consumed by land crabs when offered just one food item after they had been deprived of water for 0, 4, or 8 days. The solid symbols in the bars represent the adjusted means derived from the model coefficients and the smaller open circles are the statistical outliers (values either greater than upper or lower quartile + (1.5 * interquartile difference)). Note the different scales on the y-axis for the fresh and dry food. The data was derived from 14 different individual animals (held in separate experimental feeding chambers) for each food type and dehydration level treatment.

Table 3 Statistics for consumption of wet and dry foods.

Factor	Value	SE	t-Value	p-Value	2.5% CI	95% CI	
Intercept	4.727	0.025	191.497	<0.001	4.678	4.775	
Day 4	−0.380	0.038	−9.967	<0.001	−0.455	−0.305	
Day 8	−4.494	0.040	−12.412	<0.001	−0.573	−0.417	
Apple	0.726	0.030	24.403	<0.001	0.668	0.785	
Fish	0.576	0.031	18.767	<0.001	0.516	0.637	
Dry	−2.031	0.061	−33.335	<0.001	−2.153	−1.914	
Day 4 * apple	0.457	0.044	10.374	<0.001	0.371	0.544	
Day 8 * apple	0.467	0.046	10.119	<0.001	0.377	0.558	
Day 4 * fish	1.181	0.044	26.955	<0.001	1.095	1.267	
Day 8 * fish	0.272	0.048	5.619	<0.001	0.177	0.367	
Day 4 * dry	−0.738	0.044	−16.655	<0.001	−0.825	−0.651	
Day 8 * dry	−1.179	0.056	−20.999	<0.001	−1.290	−1.070	
Apple * dry	1.006	0.064	15.647	<0.001	0.882	1.134	
Fish * dry	−0.252	0.072	−3.478	<0.001	−0.393	−0.109	
Notes:

Food consumption of single fresh and dried food items. The following factors were included in a generalized linear Poisson regression: days dehydration (0, 4, and 8), food type (lettuce, apple, and fish), moisture level (fresh, dry) and interactions between: food type * days dehydration, days dehydration * moisture level, and moisture level * food type. The “Intercept” is the reference and represents Day 0, Lettuce, Wet—all treatments are contrasted against the reference. Residual deviance = 15,622 on 238 degrees of freedom. The data were based on 14 individual crabs for each food type and each dehydration level monitored in separate containers.

SE, standard error; Value, coefficient estimate; CI, confidence interval.

Table 4 Number of crabs feeding on wet and dry food.

Meal	Dehydration (days)	
	0	4	8	
Fresh lettuce	14	13	11	
Dry lettuce	6	5	3	
Fresh apple	13	14	13	
Dry apple	14	10	7	
Fresh fish	12	14	13	
Dry fish	9	8	2	
Note:

Number of animals feeding (total of 14) on fresh or dry lettuce, apple, or fish when offered a single choice of each item as a function of being dehydrated for 0, 4, or 8 days. The data represent a total of 14 animals (with one animal per experimental container) per food item and different animals were used for each food item and each dehydration level.

In contrast to the single food type experiments when crabs were offered multiple fresh food items, fish was consumed in higher amounts compared to apple, followed by lettuce, of which only small amounts (<8% of all food) were consumed (Fig. 6; Table 5—see coefficients for the day * food type interactions). There was also an interactive effect of dehydration; for lettuce only very low amounts were consumed and there was no effect of dehydration on the amount consumed (GLMM, df = 117, t = −1.01 and 0.96, p = 0.32 and 0.34). The crabs ate less apple at 4 days, compared to 0 day, and the amount consumed dropped further at 8 days (GLMM, df = 117, 4 days, t = −7.95, p < 0.001; 8 days, t = −11.82, p < 0.001). For fish a significant effect of dehydration was only evident at 8 days (GLMM, df = 117, t = −3.60, p < 0.001), here the 1.6% ± 0.04% BW consumed was lower than that measured at 0 day (2.1% ± 0.04% BW) and 4 days (2.4% ± 0.04% BW). These differences were underpinned by the number of animals feeding (Table 6); most of the crabs (between 10 and 13) ate some apple and fish when given a choice of all three food items, while only 5–11 individuals fed on the lettuce (Table 6).

Figure 6 Multiple choice of items offered to land crabs.

Boxplots showing amount of fresh lettuce, apple, or fish consumed (% dry mass as a function of animal dry mass) by land crabs when offered a multiple choice of all three items after they had been deprived of water for 0, 4, or 8 days. The solid symbols in the bars represent the adjusted means derived from the model coefficients and the smaller open circles are the statistical outliers (values either greater than upper or lower quartile + (1.5 * interquartile difference)). The data was derived from 14 different individual animals (each held in a separate experimental chamber) for each dehydration level treatment. All three food items were offered in excess in order to ensure crabs did not consume all of one food item and then just move onto the next item.

Table 5 Multiple choice experiment statistics.

Factor	Value	SE	t-Value	p-Value	2.5% CI	95% CI	
Intercept	2.698	0.069	38.919	<0.001	2.559	2.831	
Day 4	−0.101	0.101	−1.005	0.315	−0.299	0.096	
Day 8	0.092	0.096	0.957	0.338	−0.096	0.280	
Apple	2.700	0.072	37.690	<0.001	2.562	2.843	
Fish	2.264	0.072	36.540	<0.001	2.486	2.768	
Day 4 * apple	−0.844	0.106	−7.950	<0.001	−1.052	−0.636	
Day 8 * apple	−1.122	0.103	−11.817	<0.001	−1.413	−1.011	
Day 4 * fish	0.268	0.104	2.580	0.010	0.064	0.472	
Day 8 * fish	−0.360	0.100	−3.597	<0.001	−0.556	−0.164	
Notes:

Multiple choice experiment. The following factors were included in a generalized linear Poisson regression: days dehydration (0, 4, and 8) and food type (lettuce, apple, and fish) and interactions between: food type * days dehydration. The “Intercept” is the reference and represents Day 0, Lettuce—all treatments are contrasted against the reference. Residual deviance = 10,697 on 117 degrees of freedom. The data were based on 14 individual crabs at each dehydration level monitored in separate containers.

SE, standard error; Value, coefficient estimate; CI, confidence interval; %diff, percentage change

Table 6 Number of crabs feeding in multiple choice experiment.

Meal	0 day	4 days	8 days	
Lettuce	5	7	11	
Apple	11	10	12	
Fish	12	13	12	
Note:

Number of animals feeding (total of 14) when offered a multiple choice of fresh lettuce, apple, or fish as a function of being dehydrated for 0, 4, or 8 days. The data represent a total of 14 (different) crabs, for each dehydration level (each crab was held separately in an experimental chamber).

When a wider range of crab sizes encompassing juveniles (25 g) to adults (480 g) were included, diet preferences of crabs were found to be size dependent (Generalized least squares regression, df = 162, t = −5.60, p < 0.001; Fig. 7; Table 7). As in the other experiments, regardless of crab size apple and fish were preferred over lettuce. However, smaller crabs ate slightly more lettuce than the larger animals. In addition the smaller crabs ate almost twice as much fish as the largest crabs and the amount of fish consumed declined as the crab mass increased. In contrast, the largest crabs ate twice as much apple compared with the smallest crabs and the amount of apple consumed increased with increasing crab size (Fig. 7; Table 7; Generalized least squares regression, df = 162, t = 2.63, p = 0.01).

Figure 7 Effect of crab size on feeding preferences.

Amount of fresh lettuce, apple, or fish consumed (% dry mass as a function of animal dry mass) of land crabs varying in size between 25 and 475 g when offered a multiple choice of the three food items. Each crab was maintained it a separate experimental container and the food items were offered in excess in order to maintain a multiple-choice of food items throughout the 12 h experimental period. Only fully hydrated crabs were used in this experiment.

Table 7 Statistics for crab size and food choice.

Factor	Value	SE	t-Value	p-Value	
Intercept	35.206	4.447	7.917	<0.001	
Crab mass	−0.093	0.017	−5.590	<0.001	
Apple	90.327	35.590	2.538	0.012	
Fish	199.968	40.694	4.914	<0.001	
Crab mass * apple	0.349	0.133	2.626	0.010	
Crab mass * fish	−0.165	0.152	−1.088	0.278	
Notes:

Effect of crab size on food preference. An individual crab was held in the experimental chamber and offered the three food types. A general least-squares regression included crab body mass and food type (lettuce, apple, and fish), with an interaction term. The “Intercept” is the reference and represents “Lettuce” for crabs with 0 mass—all treatments are contrasted against the reference. A different standard deviation per food type was modeled (using a weights function as described in the methods) with a ratio of Lettuce = 1.000, Apple = 1.146, and Fish = 0.126. Degrees of freedom = 162.

SE, standard error; Value, coefficient estimate.

Discussion

Overall, dehydration affected how much G. ruricola consumed, with a significant decrease in all food items, but especially dry matter, with increasing dehydration status. Given the future predictions of drier climate for the Caribbean, the corresponding dehydration in this species will influence its ability to fulfill its role as an ecosystem engineer in coastal forest ecosystems.

Precipitation levels and responses to dehydration

A decrease in Caribbean rainfall levels of between 25% and 50% is forecast by the end of the century (Nurse & Sem, 2001; Christensen et al., 2007; Campbell et al., 2011). Given the potential loss of standing water and associated lower humidity this would increase the number of days that the crabs would be at a higher risk of dehydration stress and thus alter foraging patterns. Moreover the timing of the dry season is important, which in the Caribbean lasts from November through to April (Chen et al., 1997; Campbell et al., 2011). However, the rainfall data from south Eleuthera, Bahamas showed the greatest number of consecutive days without rain between September and January. Given this scenario it could lead to an increase in the mean number of dry days during September–January from 13.5 days to between 16.9 (25% increase) and 20.3 days (50% increase), a significant finding given that the crabs in our study became moribund after only 14 days under the predicted future climatic regime. The temperatures during the first part of this dry season are still high and the crabs would be active and foraging, rather than hibernating in burrows (Bliss, Wang & Martinez, 1966; Bliss et al., 1978) and so they would be directly affected. Although the crabs may have be able to obtain some of their water needs through metabolic water or drinking dew (Wolcott & Wolcott, 1988), this was clearly insufficient. The fact that the crabs emerged during the daylight hours, and risked predation (Ortega-Rubio et al., 1997) to drink from temporary pools indicate that precipitation events are essential in order to balance their water budget.

Gecarcinus ruricola could withstand 23% ± 2% body water loss, which is similar to the 21–22% water loss reported for the closely related species G. lateralis (Bliss, Wang & Martinez, 1966) and within the range of other land crabs (Herreid, 1969; Wood, Boutilier & Randall, 1986). Fatal body water loss occurred within 14–16 days without access to water. During this time the crabs were not fed; one would assume that metabolic water from food would be very important (Wolcott & Wolcott, 1987; Wolcott, 1992), and although 23% body water loss would likely be fatal, the time to reach this level would typically be longer than 14–16 days. It could also be argued that the crabs would retreat into the burrow where the air is usually fully saturated, and that this would slow water loss (Bliss et al., 1978). However, G. ruricola is not always able to construct or inhabit burrows and the larger animals in particular are often found in crevices or under rocks where they would be more prone to dehydration (Wolcott, 1992; Griffiths, Basma & Vega, 2007; present study observations). That being said, given the use of metabolic water and changes in behavior, even the most extreme predicted climatic changes would probably not prove fatal for this species. Nevertheless, an increase in dehydration levels coupled with changes in feeding patterns will likely lead to reduction growth and overall physiological condition in these crabs (Bliss et al., 1978; Wolcott & Wolcott, 1984).

Oxygen consumption rates of water deprived G. ruricola remained unchanged until 8 days of dehydration; because the hydrated animals did not show the same decline in oxygen consumption, the decline in oxygen consumption in crabs without access to water was associated with dehydration rather than simply being a result of food deprivation for 8 days (Ansell, 1973; Wallace, 1973). In contrast to the responses observed for G. ruricola, oxygen consumption in Cardisoma guanhumi declines within 36 h without water (Wood, Boutilier & Randall, 1986) and even slight water loss (<4%) in Ocypode quadrata causes a decrease in VO2Max (Weinstein, Full & Ahn, 1994). Both of these species are less terrestrial in habitat than G. ruricola and its responses showed it is better able to tolerate desiccation (Taylor & Davies, 1982). During experiments G. ruricola were active and could be heard moving around in the covered plastic containers, but the dehydrated animals were noticeably less active at day 8. This behavioral suppression in activity as a function of dehydration has also been reported for another species, Holthuisana quadratus (Greenaway, Bonaventura & Taylor, 1983). The fact that the MMRs of G. ruricola were unaltered after 8 days dehydration (Table 2) also suggests that it was a behavioral reduction in activity, rather than a physiologically regulated mechanism. Dehydrated crabs can gain a lot water within a few hours, with pre-treatment levels regained after 24 h (Bliss, Wang & Martinez, 1966; Wood, Boutilier & Randall, 1986). Here crabs were fully rehydrated and oxygen consumption had also returned to pre-treatment levels within 24 h. We did attempt to measure oxygen consumption during the initial stages of rehydration (2–6 h), the problem being that this time period coincided with hours of darkness. The animals became very active at dusk exhibiting a doubling of heart rate (McGaw et al., 2018), and so the increase in activity masked any changes associated with rehydration.

Feeding preferences

When presented with a choice of food the crabs preferred fish and apple and consistently consumed low quantities of lettuce. Land crabs show a strong preference for high nitrogen foods such as carrion and animal faeces and will congregate around these food items in high numbers (Wolcott & Wolcott, 1984; Wolcott & O’Connor, 1992; Linton & Greenaway, 2007). Fleshy fruits contain a high proportion of living cells that are readily digestible and are preferred over leaf litter which has a higher carbon to nitrogen ratio and higher levels of cellulose (Linton & Greenaway, 2004). Therefore it is not surprising the crabs selected fish and apple, but ate low amounts of lettuce. Erickson et al. (2008) also found that although the mangrove crab, Aratus pisonii, primarily feeds on leaves in their natural habitat these are only eaten in very low amounts when other food items are offered. This opportunistic omnivory is common in herbivores and leaves are most likely only eaten as a necessity (Erickson et al., 2008; Nordhaus, Salewski & Jennerjahn, 2011). A selective preference for the high energy food type (when offered a choice) therefore explains the low lettuce intake. However, it does not explain why a low intake also occurred when only lettuce was offered to the crabs (single choice experiments). Because lettuce leaves are nutrient limited it might have been expected that crabs would show compensatory feeding and eat more of them (Greenaway & Raghaven, 1998). The reasons for this feeding pattern are unclear. It is possible that compensatory feeding did not occur because lettuce leaves are similar to the crab’s natural diet of green leaves (which they were maintained on before experiments) and they were exhibiting a negative preference induction whereby they preferred novel items (Thacker, 1996, 1998). This has been observed in the land hermit crab Coenobita compressus, which reduce intake of familiar foods, preferring novel items that may provide them with essential nutrients and enhance growth (Thacker, 1998). An alternative explanation is that land crab preference may not be solely dependent on nutrient content or novelty, but could be based on other factors such as palatability or texture of the food (Nordhaus, Salewski & Jennerjahn, 2011).

While we expected that as crabs became deprived of water they would choose food items with a higher water content, instead crabs preferred the food with the higher energetic content, irrespective of dehydration status. The crabs also consumed less of each food item and were less likely to feed as dehydration levels increased, and this was most pronounced for the dry food items. This decrease in food intake could be due to several reasons. In dehydrated mammals a lower food intake reflects a lower metabolism (Silanikove, 1994). This is unlikely to be the case here for G. ruricola because although they exhibited a reduced oxygen consumption rate, it was only after 8 days of dehydration and this appeared to be due a reduction in activity rather than a down-regulation of metabolism. The reduced appetite in dehydrated mammals is also related to the inability to produce adequate amounts of saliva (Silanikove, 1994; Willmer, Stone & Johnston, 2005; Maloiy et al., 2008). Certainly in Gecarcinid crabs the gut plays a role in water storage (Mantel, 1968) and during dehydration water may be taken from the gut to replace that lost from the hemolymph (Harris & Kormanik, 1981). Since the foregut is the site of food processing and requires the input of gastric juices this seems a likely explanation of why the dehydrated crabs ate less food, especially dry food items (McGaw & Curtis, 2013). In addition, as crabs lose water the hemolymph osmolality increases (Harris & Kormanik, 1981). When dehydrated, crabs may eat less because digested nutrients would be transported as amino acids and glucose which would temporarily increase the osmolality of an already elevated hemolymph. The subsequent intracellular catabolism of nutrients leads to the production of nitrogenous wastes and voiding these wastes in the urine would also increase water loss (Harris, 1977). Dehydrated land crabs may suspend processing of the meal, lowering protein catabolism, and subsequent nitrogenous waste production (Wood, Boutilier & Randall, 1986). We did notice that dehydrated crabs did take longer produce faeces when dehydrated. However, this was probably only a slowing, rather than a total suspension of digestion (McGaw & Curtis, 2013). Gecarcinid crabs can tie up toxic ammonia as urate crystals, removing it from the system and thus the need to produce urine excrete it (Linton, Wright & Howe, 2017); these urate crystals can also function as a subsequent nitrogen store (Wolcott & Wolcott, 1984). However, in Cardisoma guanhumi (Wood, Boutilier & Randall, 1986) ammonia and urea levels increase over 72–84 h, before declining, suggesting that the crabs are unable to convert nitrogenous wastes to urates immediately. Thus, the decrease in amount of food consumed may be a balance between the need to gain nutrients and metabolic water coupled with inability to produce adequate gastric juices and to immediately store the nitrogenous wastes.

Gecarcinus ruricola consumed considerably less dry than fresh food, irrespective of dehydration status or food type. Cardisoma hirtipes also prefers fresh green leaves and flowers to older dryer material; however, if only dry brown leaves are available they actually eat more in order to extract more nutrients (Greenaway & Raghaven, 1998). Similar compensatory feeding was not observed here, the wet food could simply be more palatable and food choice may also be based upon texture and not just nutrient content (Nordhaus, Salewski & Jennerjahn, 2011). This low intake of dry material may have important implications for natural foraging: fresh leaves that fall and become available to the crabs dry quickly (Kellman & Delfosse, 1993), and given a future drying scenario there will likely be more dry leaf litter, but less of it being consumed by the crabs.

Finally there were differences in food preferences of non-dehydrated crabs as a function of size. Smaller juvenile crabs ate more fish, while larger adult crabs consumed more apple; in line with the other preference experiments, very little lettuce was consumed. Fleshy fruits are often selected because they are easily digested (Linton & Greenaway, 2004, 2007) coupled with a relatively high energy content the apple may provide the necessary nutrients for adult crabs. The herring had the highest protein and nitrogen content, and since small crustaceans moult more frequently it might be expected that they would require a higher protein and nitrogen intake (Hartnoll, 1988). Indeed, intermoult periods are lower and more growth likely occurs in land crabs when they are not protein and nitrogen limited (Wolcott & Wolcott, 1984).

Ecological implications

Many land crab populations in the Caribbean have already been reduced by over harvesting (Alayon, 2005; Baine et al., 2007), and the continued growth and urbanization in this region will only exacerbate the situation (Cincotta, Wisnewski & Engelman, 2000). If dehydration levels alter foraging patterns of G. ruricola, resulting in a lower food intake, this would ultimately slow growth leading to smaller, less healthy crabs (Wolcott & Wolcott, 1984). A reduction in the fishery will further impact the expanding human population, because land crabs are an important source of protein and income for many Caribbean Islanders (Baine et al., 2007). Direct human impacts due to a reduction in crab numbers may however be less severe compared to potential trickle-down effects that the loss of land crabs would have on the environment. Gecarcinid land crabs can reach densities of 10,000–60,000 per hectare (Kellman & Delfosse, 1993; Sherman, 2003); these animals have been described as ecosystem engineers because of their role in nutrient recycling and seedling recruitment (Lindquist et al., 2009). Land crabs are very important in forests because they feed upon and reduce surface leaf litter (Kellman & Delfosse, 1993; Sherman, 2003). They also bring food down into their burrows thereby enriching nutrient levels deeper in the soil (Sherman, 2006). Leaf litter rapidly builds-up in areas absent of crabs, preventing seedlings from germinating, altering soil nutrient patterns, and preventing precipitation soaking into the soil (Kellman & Delfosse, 1993; Sherman, 2003; Lindquist et al., 2009). Land crabs also prey selectively on seedlings and fruit and as such dictate the diversity of species that can become established (Green, O’Dowd & Lake, 1997; Sherman, 2002; Capistrán-Barradas, Moreno-Casasola & Defeo, 2006; Lindquist et al., 2009).

Although a reduction in precipitation levels might lead to changes in land crab foraging activity that will affect nutrient balances and floral diversity (O’Dowd & Lake, 1989; Capistrán-Barradas, Moreno-Casasola & Defeo, 2006; Lindquist et al., 2009), such ecosystem changes are unlikely to be driven by changes in crab foraging alone (Parmesan & Hanley, 2015). The decrease in available water for the plants will also play a major role in shaping coastal forests. Predictions vary as to whether there will be a shift in plant species richness, or whether plant communities will adapt to periods of drought (Engelbrecht et al., 2007). Nonetheless, the current literature suggests the predicted drying will lead to a slower growth rate, particularly in saplings, and a loss of 30–40% of plant biomass (Allen et al., 2017). The reduced rainfall will lead to a bottleneck of periods when seedlings can germinate (McLaren & McDonald, 2003), while predation by crabs will further reduce the numbers of seedlings that become established (Capistrán-Barradas, Moreno-Casasola & Defeo, 2006; Lindquist et al., 2009). A reduction in precipitation also limits the transfer of soil nutrients for plants, especially nitrogen (Allen et al., 2017); this will likely be further compounded by the reduced turnover of surface nutrients by the crabs (Sherman, 2006). Thus, there is complexity in how this ecosystem will responds to future climate change, suggesting that this system is compelling for research on species interactions and ecosystem functioning in a warmer and drier climate.

Conclusions

Black land crabs, G. ruricola could withstand a body water loss of 23% ± 2% and survive for between 13 and 16 days without access to water. The crabs consistently chose the food with the higher energetic content irrespective of dehydration status. However, an increase in dehydration levels led to a reduction in food intake in G. ruricola and this was especially noticeable for dry food. This lower food intake likely occurred because loss of water from the gut would hamper digestive processes. Land crabs are important ecosystem engineers and the predicted decrease in Caribbean rainfall could have important trickle down effects on coastal forest ecosystems.

Supplemental Information

Supplemental Information 1 Multiple choice feeding amount.

This contains multiple choice raw data and calculations embedded in the file.

Click here for additional data file.

Supplemental Information 2 Single offerings of food.

This is data for wet and dry feeding, it contains embedded calculations

Click here for additional data file.

We would like to thank the director and staff of the Cape Eleuthera Institute for their help and use of facilities. We also thank Dr. Stuart Linton, Deakin University, for helpful discussion.

Additional Information and Declarations

Competing Interests

Author Contributions

Data Availability

Amanda E. Bates is an Academic Editor for PeerJ.

Iain J. McGaw conceived and designed the experiments, performed the experiments, analyzed the data, contributed reagents/materials/analysis tools, prepared figures and/or tables, authored or reviewed drafts of the paper, approved the final draft.

Travis E. Van Leeuwen contributed reagents/materials/analysis tools, authored or reviewed drafts of the paper, approved the final draft.

Rebekah H. Trehern performed the experiments, contributed reagents/materials/analysis tools, authored or reviewed drafts of the paper, approved the final draft.

Amanda E. Bates analyzed the data, contributed reagents/materials/analysis tools, prepared figures and/or tables, authored or reviewed drafts of the paper, approved the final draft.

The following information was supplied regarding data availability:

The raw data is available as Supplemental Files.

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
