# Peer review of "Changes in precipitation may alter food preference in an ecosystem engineer, the black land crab, Gecarcinus ruricola"

_PeerJ, doi:10.7717/peerj.6818_

## Round 0.1 · original submission · Major Revisions

I now have comments back from two expert referees who each have major issues with the design of the experiment and your analyses. While both acknowledge that you have done a LOT of work here, and the experiments are likely to be informative, they also raise issues of pseudoreplication and replication issues that led one to conclude the analyses are incorrect, and prevented the other from being able to determine exactly how you performed your analyses. Between them, the referees have provided extensive feedback on the manuscript and the changes that would need to be made prior to your manuscript becoming acceptable for publication. Many of these are clarifications or additional details that could be addressed rather easily with a thorough edit and detailed revision. However the issues of experimental design and analysis raised by both are more serious and will need careful attention. While serious enough that neither referee felt the paper was publishable in its current form, neither felt that the issues were necessarily fatal if you were able to address the concerns and provide appropriate analyses for the experiments you performed.

Given both referees felt the analyses were inappropriate, your manuscript will require additional review following revision. Please make sure to include a point-by-point rebuttal to the details reviews with your revision so that the referees are able to see how you have addressed their concerns.

·

Basic reporting

1. BASIC REPORTING
- Clear, unambiguous, professional English language used throughout.
o While the writing is general strong, the authors introduce numerous moments of ambiguity and breaks in flow that are both problematic and easily fixed. I recommend a substantive, line-by-line review of the current report with renewed attention to these weak points many (most?) of which I point out marginally in the ms.
o I make multiple suggestions for potentially improved clarity and flow throughout the entire paper from abstract to figure legends. I was unable to comment on the pdf tables/figures themselves but attempt to make a few comments in the associated legends.
o There are some key issues I have related to grammar and word choice. While these issues represent significant (and sometimes critical) flaws in presentation, nearly all are easily fixed.
 E.g., The use of the term “single preference” in regards to the single-item food tests is an oxymoron. No such thing. As such, the word preference can be replaced with “food consumption amount” or some such.
 E.g., The grammatical conjunction of “lettuce, apple and fish” (with only one comma) means something entirely different than “lettuce, apple, and fish” (with two commas). In the first example, the crabs have two options: lettuce OR apple and fish. In the second, they have three.
 Little details like these two emerge throughout the entire paper including the legends and make the reader work far more than we should to develop a tenuous understanding of what it is you want to convey.
 I flag most of these moments in the document and, thus, I hope that they are easily found and fixed.

o STYLE-wise, the authors’ affinity for starting sentences with “This” or similar when “this” can refer to any one of several possible takers from the previous sentence introduces unnecessarily ambiguity. Again, “this” is easily fixed.
o Some of your sentences (marked) float in that they have neither connection to what precedes or succeeds them. Again, a few cleverly places transitional words and its fixed.
o You may consider adding more verve to your writing by replacing the verb “to be” (e.g., is, are, were) with more powerful verbs (after all, any verb is more powerful than “to be” in all but the most intentional of circumstances).

- Intro & background to show context. Literature well referenced & relevant.
o The authors generally review the literature well and use a professional tone and style to illustrate several important elements that support the research to be reported. That said, I make many comments throughout about ways to further improve this intro.

- Structure conforms to PeerJ standards, discipline norm, or improved for clarity.
o The structure is conventional and, thus, form-fit to the industry’s standards. However, there is a missing subsection of the methods that I feel needs to be added: The study system description. I explain in the margins of the paper in far more detail than here just what I feel is missing. In summary, we need to know all relevant aspects of both the crabs (addressed in moderation elsewhere in the paper and thus easily moved to this new section) and the island forest ecosystem (including climate etc.) to understand better the experimental design and results/discussion to follow. In summary, I find this to be an essential addition to the revision.

- Figures are relevant, high quality, well labelled & described.
o I would say that several of these figures are quite (perhaps, overly) complicated. I also believe that some of them are incorrectly presented. For example, I have been taught that when one finds significant interaction effects, then the individual elements of the interaction (although also statistically significant) are NOT presented as such. Once the interaction is considered significant, the individual components can no longer be considered significant in the absence of the interacting co-variable. (If a significant interaction was NOT found, then (and, I have been taught, only then) would one present the factor in isolation. (table 4)
o Additionally, I find the regression analysis to be unconvincing as we are looking at essentially flat line regressions with huge amounts of noise. My recollection about the use of p-values in regression analyses is that huge samples sizes (like they are presenting) can make even the most biologically INsignificant findings, statistically significant. For this reason, some stats programs even refuse to calculate the p-values for such regressions (even tho it is easy to do) simply because they know that such p-values can throw off well-intended researchers into making a type 1 error. I think this problem emerges within figure 7.
o I feel that for nearly all figures and tables, additional context and description in the legends would be in order.
o I also question the need for so many tables and figures as some of the information (e.g., table 1) could be easily incorporated into the text. I recall that energetic content of the food was a detail specifically missing from the results text that I felt should be included (not sure I am recalling correctly but you can check).
o I was hoping to see df for table 4 (or at least mention of sample size in the legend) for table 4 and similar. I was skeptical of the highly significant results for all factors and made me wonder about the potential likelihood of pseudo-replication (see more on this potentially serious concern below).

- Raw data supplied (see PeerJ policy).
o I cannot find the raw data (nor do I feel I need to see it). I looked for it using the links supplied but they did not lead me to the raw data. I have no opinion about this one way or the other.

Experimental design

KEY: Prompts from the editors in bold. My additions in bold italics.

2. EXPERIMENTAL DESIGN
- Original primary research within Scope of the journal.
o To the best of my knowledge, this ms complies completely with the journal’s requirement for original primary research. I am assuming that the ms falls within your journal’s scope because I was asked to review it. Sorry, I am not yet familiar with your journal so cannot speak more without more effort But I have NO yellow flags of any sort on this point.

- Research question well defined, relevant & meaningful. It is stated how the research fills an identified knowledge gap.
o Largely yes. I find the authors’ work to be impressive in its intensity and scope but somewhat unpersuasive as to its relevance to natural circumstances either present or future. My main concern is about the powerfully contrived nature of the overall design which was highly manipulative and largely if not wholly artificial. (Although the authors do make a case for using novel food items not found in nature.)
o The research imposes several hypotheticals some of which are more plausible (e.g., dehydration) or less plausible (e.g., unnatural food choices). Then, the researchers combine these distinct factors in ways that further compound the hypothetical aspects of the work. Thus, I would have found this all more persuasive if naturally available foods were used instead. I would have found it even more persuasive if field experiments were used to corroborate the lab manipulations. However, that said, I feel that such work must begin somewhere and this early effort reflects a strong early go at the topic albeit from one extreme of the natural vs controlled spectrum.
o Regarding the authors’ efforts to make a case that such research is relevant in the face of a drying and warming climate, I feel that they did a stand-up job. Well done because even a skeptic about the validity of lab manipulations—like me—found the argument somewhat compelling.

- Rigorous investigation performed to a high technical & ethical standard.
o Yes, in most ways but see below RE ethics.
o They were thorough sometimes to the extreme in their work and design and I was regularly impressed with their attention to detail in design. They were rigorous in their data gathering (both in scope and detail) and analyses.
o That said, I have a few lingering and important issues;
 Their presentation of their highly technical efforts left several key holes for their readership to fall through. As such, I frequently did not understand key elements of their methods and findings despite my dedicated attention during this review. Thus, I feel that it is the presentation of their work (rather than the work itself) most in need of revision.
• For example, understanding the correct samples sizes of the crabs in the foraging experiments was, for me, nearly impenetrable. Thus, I could not fully comprehend (nor comprehensively review) these core elements. That is just one of many such examples.
• As such, a substantive, line-by-line revision (particularly of the methods section) will markedly improve the paper’s accessibility for an interested readership. All that said, the corrections are mostly quite simple. I suspect that they mostly reviewed the work amongst one another and, thus, were unable to reveal ambiguities that jump out of the page for a first-time reader. As one of these first-time readers, I was (I have to admit) rather reeling from the number of unanswered questions I had about process, purpose, design, etc. Again, all of this is easily (and essentially) repairable.
 Regarding ethics, they suggest that they are in compliance with some oversight effort but we cannot deny that the research amounted to a catastrophically difficult time for many of the crabs and, to be transparent, I admit that I struggled, personally, with this. So far as I can tell, 200 or more crabs (again, this part is unclear so I cannot be sure) were subjected to capture and captivation in unnatural conditions (e.g., wire cages). A subset of those individuals (again, unclear as to how many but over 150, I think) were subjected to various levels of dehydration. A subset of those were subject to levels of dehydration to the point that they became unresponsive (but revivable, I think). And, 18 crabs (pretty sure) were placed in ice water to kill them so that they could be measured for dry weights and for use in what I felt was a somewhat non-essential efforts to calculate consumption amounts as a function of body weight. I don’t feel that they well defended the need to gather dry body weight of the crabs as essential.
• They might have killed one crab at a time to calculate wet wt – dry wt until the graph began to plateau at which point they would get the accurate value they sought, while, potentially, killing far fewer crabs in the process.
• I think the collection of feces from the experimental subjects as they became increasing dehydrated could have revealed important physiological insights to their stress. Testing the feces for moisture (wet wt – dry wt) might have been useful.
• So, in confess, I was personally troubled by some of the methods used and not well defended because of their lethality. That said, they claim to have been in compliance and I do not doubt them. (I just don’t find the laws of compliance all that worthwhile.)

- Methods described with sufficient detail & information to replicate.
o No. I feel that the report’s greatest weakness lies with the sheer number of points of critical ambiguity inadvertently built in (or left out) of the text. I have, perhaps, 50 or so questions about design details, data presentation details etc., which, in this current draft, remain unanswered or unclear (for me).
 The bad news is that I feel that without great clarification of these essential elements that pepper a majority of the ms, this paper should not be published.
 The good news, is that all of these concerns (well, most of them as some of them cut to the design itself) can be easily fixed through additional information, clarification of the information already provided, and, in some cases, proper use of grammar.
o I suggest that they go line-by-line through the methods section and address all the questions that arise during/after a first-time reading of the materials.
o That said, they had a huge amount of information provided and the authors should be commended for presenting a complex design and analysis in great detail. However, given the complexity of their work, even more (far more) details are needed.

- IMPORTANT ISSUE (NOT PROMPTED): Potential design flaws.
o I am skeptical of the design of the food consumption trials (erroneously called “preference” trials) as well as the actual food preference trials (three types of food offered). In particular, I think we need to know how/when/if the food replenished during the trials. Was the food replenished regularly? Was the food allowed to desiccate over time? What sort of behavioral ecology was going on when males, females, juveniles and adults were mixed (i.e., did the juveniles or females eat less preferred foods due to the dominance of the larger adult males)? In other words, I can imagine (but do not now know due to the incomplete description of the methods) potentially fatal flaws to this design that would render it unusable at worst or in need of major moderation of the interpretation of the results at best.
 For example, in the three-food trials, findings revealed that lettuce was not preferred compared to the apples and fish, and apples were eaten a lot until about day 4 of dehydration. Then apple eating dropped off. Q: Did they eat fewer apples because there was no more apple left? Because their deteriorating condition led to a switch to fish from apple? Because the apples became desiccated? These details that (so far as I recall) are missing from the methods description, radically alter the interpretations of the findings.
o Pseudo-replication? When 14 crabs are placed into a bucket with a food source from which they eat or don’t, can we really reference this trial as having a sample size of 14? I don’t think so lest we engage in pseudo-replication (of Stuart Hurlbert fame). Rather, because the crabs cannot be assumed to be acting independently of one another when confined in close circumstances and feeding from the same sources and, thus, they violate the assumptions of the statistical tests later performed. As such, it appears to me that the sample size of that particular trial is 1 (for which there were 14 crabs involved). Therefore, this potential problem is far less easily fixed than the ones mentioned above and I think this should be further examined but this design (albeit not sufficiently well described in this current draft) feels like a classic case of pseudo-replication. Perhaps the other reviewers caught this too?

Validity of the findings

3. VALIDITY OF THE FINDINGS
- Impact and novelty not assessed.
o (I don’t understand the negative angle of this prompt.) So, let me just say that the authors stated (inferred) well the potential impact and novelty (to a lesser degree) of their findings. If anything, they pushed the limits of this inference given the highly controlled and unnatural design they created. Thus, I feel this work is a good early step to assess the potential relationships between dehydration regimes and novel food consumption patterns. So, if anything, I might tone down the inferences a bit, but I don’t think this is an issue that floats to the top for their next revision.
o See comments in section immediately above about potential design problems and analysis problems.

- Negative/inconclusive results accepted.
o I don’t find persuasive their conclusions illustrated by figure 7 that presumably correlates food “preference” (poorly chosen term for a single-diet option treatment) and crab size (dry weight). I find these data to reveal absolutely NO relationship between the two variables yet the authors (citing the statistically significant results that I am claiming to be a possibly spurious artefact of the large sample sizes and their effect on the regression). I want to be clear that the authors seem to be more knowledgeable about statistics than I am so my confidence is lower than otherwise. But, I believe that they are making a Type 1 error here in which they are claiming a significant outcome where none exists (not biologically significant at any rate). If I am wrong, forgive this intrusion.
o See my marginal comments for other issues.

- Meaningful replication encouraged where rationale & benefit to literature is clearly
stated.
o Again, this prompt confuses me a bit but their replication was, in general excellent so long as they did not engage in pseudo replication. See my comments above regarding pseudoreplication.

- Data [are] robust, statistically sound, & controlled
o See above for concerns about pseudoreplication and lingering questions related to their food availability regimes (not described). All of this is mentioned above in the section called “important issue”.

- Speculation is welcome, but should be identified as such.
o The authors are solid on this. They are generally modest in the phraseology of their extrapolations and assertions but bold in their desire to make them in the first place. I find this to be a strength of the paper as it makes for provocative reading.

- Conclusions are well stated, linked to original research question & limited to supporting results.
o Generally, yes. Except they bring up energetic content of the food which, so far as I could tell, was barely raised in the ms and sometimes equated with “nutrients” but without (as I recall) defense as nutrients and energetics (caloric content) are distinct so far as I know. This should be checked and, if I am correct, easily fixed from abstract to legends.

Additional comments

Review of Changes in precipitation may alter food preference in an ecosystem engineer: the black land crab, Gecarcinus ruricola written by Iain J. McGaw,Travis E. Van Leeuwen, Rebekah H. Trehern, and Amanda E. Bates.

Reviewer: Peter Sherman, Prescott College, Arizona.

This study attempts to predict how individuals of a population of Caribbean black land crabs might respond to predicted future climatic changes with a focus on decreased rainfall and/or humidity. The authors hypothesize that food consumption rates and food type choice might be altered by increasing stress associated with advancing dehydration. To test their hypothesis, the authors create a serious of experiments that both dehydrate (or not) individual crabs while offering the crabs three different types of novel foods that differ in water, energetic, and nutrient contents. While the authors choice to provide the crabs with unnatural food items reduced markedly the potential to infer about presumed future ecological events, this study may provide an early step at addressing the larger issue. The authors claim that given the potential for land crabs to behave as ecosystem engineers, such alternations to their ecology, in general, and foraging behaviors, more specifically, might add additional impacts to forests anticipated to be experiencing their own ecological responses to pending drought stress.

The authors deserve recognition for that sheer amount of work that this document represents and their overall attention to detail in their design. I commend their creative use of triangulation (of methods) and their fearless and interdisciplinary approach to tackling the provocative problem. While I find the work to be impressive in both its scope and depth, I argue that the current draft of the manuscript to be sufficiently in need of additional work that I strongly recommend a massive revision prior to reconsideration. Given the number of issues that I raise marginally in the document as well as below, I suspect that their substantive revision warrants additional professional review to make sure that all needed (and many recommended) modifications are made. I would not accept the paper for publication unless the revisions are made and accompanied by a highly detailed, comprehensive point-by-point (for the major points) description of how each point was addressed (either in compliance or disagreement). Importantly, many (but not all) of my critiques can be addressed rather easily. All of them, I think can be addressed—none of them are, to the best of my knowledge, fatal.

Please know that despite the critical and rather comprehensive nature of my review, I acknowledge that I might have missed several things (e.g., some of the statistics I did not understand) and many of my comments might have implied some missing information that the authors had provided (but I somehow missed or forgot etc.) All that said, however, I did give this my full attention for many hours and, thus, I think it would be wise for you to consider that if I am genuinely confused about something, that perhaps you (as the authors) might find a way to clarify the work even more than it already is. As the reader, I am prepared to do my fair share of the work to understand your work. But, I feel, that ultimately and generally the onus lies with the authors to go to whatever extend necessary to maximize the clarity of their prose. And so, I provide you this critique with the greatest of respect and empathy for how it feels to have to go back and revise the document one more time. Good luck with this. I look forward to seeing it published.

- General comments
o On the one hand, this is an impressive paper crafted by impressively trained authors who have provided a highly sophisticated and detailed account of a provocative research topic. The sheer breadth and depth of their efforts are impressive and should be commended. The detailed statistical analyses (some of which extend beyond my own training) appeared to be robust and well developed. The work, although highly contrived, provides an early understanding of how land-based, marine invertebrates might adapt (or not) to a drying climate. All in all, an impressive effort worthy, eventually, of publication.
o However, many substantive omissions of key details hamper my ability to fully understand (and, thus, review) the work. While most of these are fixable with another substantially altered draft (or more) following my many marginal prompts for additional clarity and accuracy, I worry about some design elements (mentioned above) that might be more problematic.
o Regarding potential pseudoreplication, even if the n = 14 crabs trial is reduced to a n = 1 scenario due to the inter-dependence of the crabs foraging together in a single container over time, I think that they can fix this by reanalyzing the data and revising the findings accordingly.
o Even if the data emerge to be statistically insignificant under that dire reduction in sample size, the biologically meaningful information remains the same (e.g., the crabs don’t like the lettuce as much as the apples etc.). So, they lose much of their statistical prowess but they maintain the main message that, when foraging in a forced group setting, less lettuce is eaten than similarly forced groups of crabs eating, say, apples.
o For the real preference trials of three food choices in one tank, then the sample size issue become even more confounding as the individual crab’s foraging choices are absolutely influenced by one another and the very notion of independence in this scenario is ecologically untenable especially if the forage was not replenished every day (or more) or provided ad libitum.
o And so, in summary, the paper requires a major revision of which a vast majority of the many need easily fixed. And, for those more elemental issues raised just above, the merits of the paper to be published depends, in large part, on their responses to these critiques. However, even under the worst case scenario where my concerns are well-founded, the authors might still be able to redeem the situation by simply modifying their analyses (redoing or eliminating many of them) but allowing the underlying (more important) biology to remain up front and center as these potential concerns don’t so strongly impact the biological findings as the statistical ones.

Reviewer 2 ·

Basic reporting

Line 153: Give species names
Line 249: This is results not methods.
Line 348-351: I had to read this a few times to understand what it was saying. Clarify.
Line 354: SD? SE? Clarify throughout, e.g. in Fig 3

Experimental design

The only time repeated measure is OK is if time is not expected to have an effect (which it clearly is here as the crabs will get more dehydrated over time). When that’s not expected to have an effect, the analyst can include time as a factor to check that there is no significant interaction between the factor of interest (in this case food type) and the repeated measures. But here the authors are specifically interested in time so they couldn’t do that. So they needed to either (a) include crab as a random factor (which would require a lot more measurement of each individual at each time period and doesn’t really solve the problem anyway) or (b) have different individuals for each food x time treatment (which would require a lot more crabs). I think the authors have done (a; Line 289-292), but I don’t really follow how “each crab potentially feeding on the 3 different food items” creates replication, so need more details before I could give a blessing. If it doesn’t work, then their results are uninterpretable. :( In which case the only thing they could do with the data they have is to only look at the start and end results, subtracting 8 d from 0 d for each individual and then having a single data point per crab that then gets used in analyses just looking at food type. But I’m getting ahead of myself, since I don’t follow what they’ve done….
Line 158: Why sex them if they were then randomly allocated? If sex not mentioned elsewhere, just leave out.
Line 183: And because researchers are asleep? Also, I understand that respiration experiments are difficult to do when the critters are all active, which is why night doesn’t work, but you don’t actually say that, which would be confusing for readers not familiar with these sorts of experiments. So make that link, if that is what you mean by avoiding the night time experiments.
Line 212-217: Those crabs: http://2.bp.blogspot.com/-7hAXgapnsg0/U9mYRThh2ZI/AAAAAAAACnU/KQATCUiR3eQ/s1600/andy.jpg. Also, do you mean goad, rather than agitate?
Line 235: I think this a long bow to draw, there’s more to the food items that them being new to the critters and the water content.

Validity of the findings

Line 170: So we don't know that's it's just water loss? It's just change in mass, right? I think the text should be written throughout to reflect this nuance. I don't think it really affects the conclusions, but better to be rigorous in saying what you mean throughout.
Line 318-340: This is just a big slab of qualitative and anecdotal information backed up by a single photo. Not sure what to do with this as I can’t just accept what’s being said. So this slice of natural history might as well not even be in the paper. :/

Additional comments

Line 160: Except the eight that you dehydrated until they became moribund (thanks for giving me the chance to learn and use a new word!) at 14-16 days. Or is this just a funny way of saying that there aren't any Canadian and Bahamian care protocols for crustaceans?

---

## Round 0.2 · Minor Revisions

Overall, the referee and I are largely satisfied by the responses in the rebuttal letter, but neither feel those responses have yet made it into the paper itself. At a glance, the responses are adequate, but the referee points out that the revisions to the manuscript are rather modest and often fall short of the clarification provided in the rebuttal letter. I have to agree - the issues raised by the referee with regard to pseudoreplication, artificial design, incorrect analyses, etc. are likely to occur in future readers as well due to lack of clarity in the manuscript. Readers should not have to dig through your rebuttal letter to find necessary clarification or additional information, because we know few if any will – they will simply discount your work and the journal that published it. Rather than rely on the reader to interpret what you did, it is the responsibility of the authors to communicate clearly such that future misinterpretations as pointed out by the referee are avoided.

I am therefore returning your manuscript for minor revisions so that you can consider your line-by-line responses in the rebuttal letter and ensure that the revised manuscript reflects the same thought and clarity for future readers as has gone into your rebuttal letter.

---

## Round 0.3 · accepted · Accept

Thank you for clarifying your responses in the manuscript and for including the tracked-changes manuscript so that we can see where the suggested revisions were made to the text.

Having read through your revised manuscript, and with the referee being in agreement that your responses in the rebuttal letter were satisfactory, I now deem that the concerns of the reviewers have been addressed sufficiently to move your manuscript into production.

I look forward to seeing the paper come out.

#